# Zinc Plant Uptake as Result of Edaphic Factors Acting

**DOI:** 10.3390/plants10112496

**Published:** 2021-11-18

**Authors:** Vyacheslav Sergeevich Anisimov, Lydia Nikolaevna Anisimova, Andrey Ivanovich Sanzharov

**Affiliations:** Russian Institute of Radiology and Agroecology 1, Kievskoe sh., 109th km, Kaluga Region, 249032 Obninsk, Russia; lanisimovan@list.ru (L.N.A.); ais_55@mail.ru (A.I.S.)

**Keywords:** zinc, ^65^Zn, soil, soil solution, barley, lability, specific activity, potential buffer capacity, forms, labile zinc pool

## Abstract

The influence of soil characteristics on the lability and bioavailability of zinc at both background and phytotoxic concentrations in Albic Retisol soil (Loamic, Ochric) was studied using various methods. Ranges of insufficient, non-phytotoxic, and phytotoxic zinc concentrations in soil solutions were established in an experiment with an aqueous barley culture. It was experimentally revealed that for a wide range of non-toxic concentrations of Zn in the soil corresponding to the indicative type of plant response, there was constancy of the concentration ratio (*CR*) and concentration factor (*CF*) migration parameters. As a result, a new method for assessing the buffer capacity of soils with respect to Zn (*PBC*_Zn_) is proposed. The transformation processes of the chemical forms and root uptake of native (natural) zinc contained in the Albic Retisol (Loamic, Ochric) through the aqueous culture of barley were studied using a cyclic lysimetric installation and radioactive ^65^Zn tracer. The distribution patterns of Zn(^65^Zn) between different forms (chemical fractions) in the soil were established using the sequential fractionation scheme of BCR. The coefficients of distribution and concentration factors of natural Zn and ^65^Zn, as well as accumulation and removal of the metal by plants were estimated. The values of the enrichment factor of natural (stable) Zn contained in sequentially extracted chemical fractions with the ^65^Zn radioisotope were determined and the amount of the pool of labile zinc compounds in the studied soil was calculated.

## 1. Introduction

Increases in the concentrations of heavy metals (HMs) in soils as a result of technogenic pollution lead to negative effects in agricultural ecosystems such as crop losses, deterioration in the quality of agricultural products, and decreases in soil microbiological activity. However, among a wide range of pollutants, Zn deserves special attention for a number of reasons

First of all, zinc is one of the 17 elements necessary for plants, with an average content of 0.002% (20 ppm) in dry vegetative mass [1]. At the same time, it is also an important trace element—zinc deficiency negatively affects the growth and maturation of plants [1,2], which leads to crop losses and even, in the most severe cases, plant death. Zinc deficiency is often caused not only by the low metal content in the soil but also by the influence of the type of soil that determines its availability to plants. [2,3]. Thus, it was noted in [4] that Zn deficiency is observed in all soils with a low availability of Zn, which primarily include calcareous and highly phosphated soils with high pH.

In general, about 50% of the world’s soils contain insufficient amounts of Zn [5]. Zn deficiency has been observed in significant amounts of soil in Bangladesh, Brazil, Pakistan, the Philippines, Sudan, sub-Saharan Africa, India, Turkey, Western Australia, and China, as well as on the Great Plains and in the western regions of the United States [2,6].

At the same time, zinc is the most common among heavy metals (HMs) polluting soil as a result of anthropogenic impact [1,7,8,9,10]. The most significant sources of pollution are the use of fertilizers and sewage sludge in agricultural practice, emissions and discharges from the mining and metallurgical industries, and road transport [11,12]. At high concentrations of zinc in the soil, various cytotoxic effects are manifested and lead to decreases in the biomass and yield of agricultural plants.

Thus, the study of zinc behavior in soils is of particular interest. In addition, due to its high mobility in soils and lack of ability to change the degree of oxidation in the environment, Zn is a convenient object for studying HM migration.

Zn’s mobility in soils and availability to plants strongly depend on the ratio of the different forms (chemical fractions) of the element in soils [5,8,12,13].

To date, a large number of methods for determining the fractional composition of HMs in soils have been developed. All of them use specific reagents to isolate individual groups of HM compounds in soils (exchangeable, easily soluble, and associated with organo–mineral complex compounds, organic matter, carbonates, Fe, Mn oxides, etc.). Using schemes of the sequential chemical fractionation of HMs [4,10,14,15,16,17,18,19,20,21,22,23], it is possible to assess their lability and bioavailability, the distribution by fractions of the main groups of compounds and minerals in soils (exchange associated with oxides of Al, Fe, Mn, carbonates, phosphates, sulfides, organic matter, and crystal matrixes of soil minerals), the lability and bioavailability of HMs under changing soil conditions (pH, redox potential, humidity, salinity, etc.).

However, all of the above methods only allow researchers to solve the main task with a modest degree of approximation—to estimate the real pool of labile and bioavailable HM compounds. In fact, a more accurate solution of this problem is only possible when using the isotopic dilution method [11,23,24,25,26,27,28].

This method is based on the law of ideal isotope exchange [29] of ions of applied radioactive or stable isotopes acting as tracers and ions of native (stable) isotopes of the HMs under study in a soil–soil solution–plant system. The isotopic dilution method allows one to calculate the amount of reserves (pools) in the soil: (a) of the total number of labile compounds of the studied HM (“*E*-value” (*E*_HM_)) and (b) the total number of biologically available compounds of HM (“*L*-value” (*L*_HM_)).

Additionally, the isotopic dilution method, being a source of valuable data on the total amount of stocks (pools) of labile and biologically available HM compounds in soil, does not allow for the obtainment of information regarding potential lability in a soil–soil solution system and the bioavailability of various forms of HMs. To solve the abovementioned problem, it is necessary to apply an integrated approach, including the joint use of methods of isotopic dilution and subsequent fractionation of soil into which a stable or radioactive HM tracer has been previously introduced, followed by the analysis of isotope ratios for individual chemical fractions.

An important aspect of the problem of interaction of technogenic HMs (including Zn) with soils is the latter’s potential for the specific and non-specific sorption of heavy metals due to the presence of soil minerals, the high molecular weight organic compounds, occluding processes, co-deposition, and the action of other mechanisms. These processes lead to the immobilization of HMs in soils. Their total effect, which can be quantified, is manifested in the form of the buffer capacity of soils in relation to HMs [16,30,31,32].

The objectives of this work were:

(1) To clarify the issue with different models of plant behavior in conditions of Zn pollution and to evaluate the buffer capacity of the soil with respect to Zn using barley as a test plant.

(2) Determine the pool of labile compounds of native Zn using ^65^Zn as a radioactive tracer and assess the potential lability and bioavailability of individual forms of the metal in the soil under conditions as close as possible to equilibrium. For this purpose, a special flow lysimetric installation of cyclic action was developed and tested in practice.

## 2. Results and Discussion

### 2.1. Experiment I

The studied Albic Retisol (Loamic, Ochric) soil was found to be coarsely textured, poor in available potassium, weakly acidic, contained a small amount of organic carbon, and had a low cation exchange capacity. However, the content of biologically available phosphorus was found to be high (the consequences of the abundant phosphating of agricultural land during the Soviet era). As such, it was an infertile low buffer soil that was not located in the best natural conditions.

More interesting is the behavior of zinc—one of the most important trace elements for living organisms and the most common pollutant –when it enters soil in increased quantities [33].

Before the experiment, a total content of zinc in the soil were determined in the samples (Table 1).Because the soil did not contain carbonates, the results obtained for the exchange of Ca^2+^ and Mg^2+^ (5.20 ± 0.06 and 0.40 ± 0.09 cmol(+) kg^−1^, respectively) with the use of AAB-4.8 as an extractant did not significantly differ from the results obtained using neutral salts as extractants: 1 M NH_4_Cl (5.70 ± 0.68 and 0.53 ± 0.05 cmol(+) kg^−1^) and AAB-7.0 (5.21 ± 0.08 and 0.47 ± 0.10 cmol(+) kg^−1^). Therefore, extraction with AAB-4.8 was used to estimate the total pool of exchange (non-specific bound) forms of alkali, alkaline earth elements and zinc. In addition, the exchange forms of HMs extracted by AAB-4.8 from soils in the extracting solution were capable of forming weak complexes of HM(Ac)_n_ that prevented the hydrolysis and re-deposition of metals [32].

Extraction using the same reagent to obtain non-specifically related compounds of various macro- and microelements was convenient in practical terms and created conditions for the in-depth study of the relationship between links in the soil–soil solution–plant migration chain. In addition, the validity of the determination of non-specifically bound (exchange) forms of HMs in soils by extracting them with neutral salt solutions at pH 7.0 is doubtful due to the tendency of multicharged transition element cations to hydrolyze with repeated precipitation [32,34].

The mass fraction of “labile” forms of zinc in the studied soil linearly increased with the amount of metal introduced (Figure 1). At the same time, the relative content of the labile form of Zn with the amount of metal introduced into the soil non-linearly increased in accordance with a power dependence. Thus, in the native soil, the proportion of “labile (or accessible to plants)” Zn from the total metal content was equal to 33%. With increases in the dose of Zn introduced into the soil, the proportion of its “labile” forms from the total amount of metal in the soil increased to 75%.

The study of the features of Zn migration in the soil–barley system under the conditions of a vegetative experiment with increasing amounts of introduced Zn is important for the purposes of predicting the behavior of metals in agroecosystems, e.g., in the case of technogenic contamination with metal in a water-soluble form. With the help of such experiments, it was possible for us to obtain a general picture of the concentration dependence of [Zn]_plant_ = f[Zn]_soil_ without investigating the essence of the involved migration mechanisms. In our case, the obtained dependence of zinc accumulation in barley straw on the total amount of metal in the soil presented the form of a straight line in a wide range of metal concentrations in the soil from 38 to 538 mg kg^−1^, with a proportionality coefficient, called the “Concentration Ratio (*CR*)”, equal to 3.8 (Figure 2a). This indicator depended on both the properties of the soil and the individual characteristics of zinc uptake by plants.

So why does the [Zn]_plant_ = f[Zn]_soil_ dependence present a straightforward character? This was firstly due to the rectilinear nature of the dependence [Zn]_soil,labile_ = f[Zn]_soil,total_, (R^2^ = 0.98) (Figure 1) and secondly to the universal nature of the biological mechanisms of Zn absorption involved in a wide range of non-toxic metal concentrations in the soil. The relationship between the accumulation level Zn (and any other HMs) in the plant and its content in the soil can be considered a response of the plant. If there is a directly proportional relationship between the metal content in the soil and its accumulation in the biomass of plants, then this type of response is called “indicative” [35]. Attention is drawn to the fact that the concentration range of zinc is large, even in the low-fertility and low-buffer soil under study.

Figure 2b shows the dependence of the concentration of Zn in barley straw on the content of metal in the labile form in the soil. This relationship was found to be directly proportional, although slightly less pronounced than the previous case (CR_Zn,labile_ = 5.4, R^2^ = 0.96). Studying the dependence of [Zn]_plant_ = f[Zn]_soil,labile_ allowed us to more objectively assess the contribution of other soil characteristics (such as pH and cationic composition) to the migration ability of HMs based only on the content of labile form of the metal in different soils. This was due to the fact that part of labile form (α) of the total soil metal content, as shown in the [Zn]_soil,labile_ = α[Zn]_soil,total_ expression, can differ for different soils many times with the same amount of [Zn]_soil,total_ and, accordingly, make an additional contribution to the overall variance of the [Zn]_plant_ dependent variable.

The [Zn]_grain_ = f[Zn]_soil,total_ and [Zn]_grain_ = f[Zn]_soil,labile_ dependencies presented, respectively, in Figure 3a,b, were obviously non-linear. They were satisfactorily approximated by a power function. The presence of such a dependence indicates the involvement of different mechanisms during the translocation of Zn from vegetative mass into the economically valuable part of the crop—grain. In addition, at a 250 mg kg^−1^ dose of introduced zinc (Zn_250_), grain formation did not occur, and at a dose of Zn_500_, even the development of generative organs was not observed and the plants themselves did not survive to the phase of the beginning of earing.

The concentration dependence between the amount of the labile form of Zn in the soil and the content of Zn in the phytomass of barley was most clearly demonstrated by the data obtained for relatively young 21-day-old plants (Figure 4a,b). The dependences were found to be directly proportional, and their angular coefficients (*CR*_Zn_) were equal to 4.02 and 5.68, respectively; although these were close to the values shown in Figure 2 for straw, they were still 5–10% more, which indicated a slight decrease in the concentration of Zn in the straw compared to the dry biomass of the 21-day-old plants. Perhaps this was due to the effect of the “biological dilution” of Zn in the vegetative mass of barley in the phenophases following the tillering phase (21 days) and with leaf litter at maturity.

Knowing the patterns of Zn behavior in the soil–plant system in a wide range of concentrations provided us to an effective tool for predicting the accumulation of metal in agricultural products under conditions of technogenic pollution. However, without a detailed study of the liquid phase of the soil–soil solution, knowledge of the role of various mechanisms in the accumulation of HMs by plants will be incomplete. Thus, due to the almost ubiquitous increased content of Zn in soil due to technogenesis, the “zero” point (without introducing Zn) in experiments with native agricultural soils will most likely be in the area of sufficient concentrations with an indicative type of plant response. In this case, it will not be possible to establish the patterns of Zn behavior in low concentrations zones, where the accumulative type of plant response is formed. Nutrient solutions representing an aqueous extract from the corresponding soils are ideal for clarifying migration patterns in areas of low concentrations of Zn in long-term vegetation experiments.

### 2.2. Experiment II

Vegetation experiments with a water culture of barley were carried out for a fairly long period of time—21 days. As a nutrient solution, a soil solution extracted from the studied soil was used with a narrow soil:solution ratio = 1:2. Increasing amounts of Zn(NO_3_)_2_ were added to various soil solution batches. No additional nutritious elements were added to the solution, except for a small amount of nitrogen in the form of Ca(NO_3_)_2_ (based on 0.2 mM N (NO_3_^−^)) due to the need to adjust the nitrate content according to each variant with a Zn concentration of 114 μmol L^−1^ (the metal concentration of 430 μmol L^−1^ turned out to be highly phytotoxic). The obtained results regarding the accumulation of zinc in plant roots and phytomass are presented in the form of a diagram in Figure 5a.

It is obvious that with a relatively low concentration of zinc in the soil solution (<5.84 × 10^−3^ mM), an accumulative type of plant response to the metal content was observed, characteristic of the lack of an element [35]. With a similar type of plant response, the kinetic curve of metal accumulation by roots from a nutrient solution (obtained using the water culture method) was satisfactorily described by a function resembling the power-law y = a × x^1/2^ in appearance but representing a superposition of two functions: asymptotic, described by the Michaelis–Menten enzymatic catalysis equation [36], and linear [6,37,38]. Moreover, the role of the latter was found to increase with increases in the concentration of Zn in the solution. This indicated the dominance of at least two transmembrane transfer systems in the region of very low and low concentrations of Zn in the substrate, differing in the degree of affinity to metal and using both highly specific carrier proteins with respect to Zn, mainly ZIP family, and other less specific ones, the number of which is limited in the cell [1,37,39]. Due to its participation in the process of the transmembrane transfer of metal into the roots of highly specific carrier proteins, we found an increased accumulation of Zn compared to the amount of metal that the plant would be able to assimilate as a result of electrochemical diffusion processes. The linear component was primarily due to the absorption of metal ions by plant apoplast and electrochemical diffusion [37,40].

Corresponding to the specified range of Zn concentrations in the soil solution, the ranges of its concentrations in barley vegetative and root phytomass were found to be 1.7 ÷ 6.3 and 6 ÷ 25 mmol kg^−1^ of dry mass, respectively.

At higher concentrations of Zn in a soil solution, in the range from 5.84 × 10^−3^ to 1.14 × 10^−1^ mM, the processes of the non-specific transmembrane transport of Zn ions entering the root surface begin to prevail as a result of a convective moisture flow and electrochemical diffusion processes [1,7,40] and the absorption of metal ions by apoplast. In our case, it can be argued that this range coincided with the range of non-toxic (normal, or optimal [35,41]) concentrations of Zn in the soil solution. As already mentioned, this type of plant response is called “indicative” (there is a directly proportional relationship between the metal content in the substrate and its accumulation in the biomass of plants). Corresponding to the specified range of Zn concentrations in the soil solution, the ranges of its concentrations in the VPs and roots of barley were 6.3 ÷ 52.9 and 25 ÷ 160 mmol kg^−1^ of dry mass, respectively, and the values of the corresponding constant coefficients of proportionality, called “Concentration Factors (*CF*)”, were equal to *CF_Zn,roots_* = 2375 and *CF_Zn,_*_VPs_ = 640 L kg^−1^, respectively (*CF_Zn_ =* [Zn]_plant_/[Zn]_soil solution_) (Figure 5b).

Finally, in the area of high concentrations of Zn in the soil solution (1.14 × 10^−1^ mM ÷ 4.30 × 10^−1^ mM), the phytotoxicity of Zn for barley was manifested. In this case, the physiological mechanisms regulating the uptake and translocation of Zn in plants were seriously disrupted. The type of plant response could be characterized as barrier restrictive [35]. At the same time, there was a strong oppression of barley plants and their premature death.

Accordingly, it should be noted that the abovementioned indicative type of plant response [35] in relation to zinc is the most common in real conditions with man-made soil contamination via metal.

### 2.3. Assessment of Zn Mobility in the Soil–Plant System and Determination of the Inactivating (Buffering) Ability of the Soil in Case of Contamination

As already mentioned, the migration ability of Zn in a soil–plant system is influenced by biological factors related to the physiological characteristics of plants [8,39,41,42,43,44,45], as well as the edaphic factors that determine the inactivating ability or buffering of soils in relation to Zn and other pollutants [34,35,40].

In the case of an indicative type of plant response, for example, for the aboveground biomass of 21-day-old barley plants (Figure 4b), a simple ratio will be observed:*d*[Zn]_plant_/*d*[Zn]_exch_ = [Zn]_plant_/[Zn]_exch_ = *CR*_Zn,exch_ = const(1)
and (Figure 5)
*d*[Zn]_plant_/*d*[Zn]_soil solution_ = [Zn]_plant_/[Zn]_soil solution_ = *CF*_Zn_ = const(2)
where [Zn]_plant_ is the mass (or molar) fraction of HMs in the dry aboveground biomass of the plant and *CR*_Zn,labile_ is the concentration ratio of HMs in the plant in terms of the mass (or molar) content of its “labile (available to plants)” form in the soil. As follows from Figure 4b, the value of *CR*_Zn,exch_ was equal to the tangent of the slope angle of the linear section of the above dependence: 5.68.

Earlier [34,46], we proposed a methodological approach for assessing the inactivating ability of soils with respect to HMs in conditions of technogenic pollution by using test plants. It was based on the fact that plants are essentially a universal integrating link in the soil–plant migration chains of different HMs. First of all, they uptake these metals for a long time and from a sufficiently large soil volume; secondly, they demonstrate similar patterns of behavior in conditions of HM pollution. The applied pollutant was zinc. Since then, this approach has undergone some corrections and additions. Its current version is presented below.

According to the definition given in [31] (p. 34), “the buffering of the system of compounds of microelements of the soil horizon in relation to any chemical element is understood as the ability to maintain the level of concentration of the element in the soil solution of this horizon at a constant level when the element level changes from the outside.”

As a measure of the potential buffering capacity (*PBC*) of soils in relation to a chemical element (for example, potassium), F. Beckett [47] proposed the use of the quotient between the change in the concentration of an exchange-sorbed element in the soil (*Q*, capacity factor) and the activity of the cations of the element under study in a quasi-equilibrium extracting (soil) solution *a*_Me_, normalized for the total activity of macronutrient cations *a*_Ca+Mg_ (*I*, intensity factor) in a wide range of concentrations of the element in the soil, with the exception of extremely low concentrations at which deviations from the law of ion exchange are observed. In a formal form, a similar dependence for Zn^2+^ ions can be written as the following equation:(3)PBCZn=QI=Δ[Zn]exchΔAR
where Δ[Zn]_exch_ is the amount of metal in the exchangeably sorbed state that the soil absorbed from the equilibration solution or, conversely, gave into the equilibration solution in comparison to the initial content of exchangeably sorbed Zn in the soil; Δ*AR* is the ratio of the activities of the cations of the metal under study (*a*_Zn_) and the cations of the macroelements Ca^2+^ and Mg^2+^ (*a*_Ca+Mg_) in a quasi–equilibrium soil solution at different concentrations of Zn in this solution. In the graphical representation of the dependence Δ*AR* − Δ[Zn]_exch_, the tangent of the slope angle of the linear part of the obtained dependence characterizes *PBC*_Zn_.

Equation (3) follows from the law of acting masses for exchanging cations in the system cation exchange complex (CEC)—a quasi-equilibrium soil solution. Thus, the equivalent exchange of divalent zinc cations on negatively charged surfaces of soil particles for Ca^2+^ and Mg^2+^ cations (which in a sum of about 90% of the exchange positions in CEC [48]) can be expressed using the following exchange reaction equation:[Ca + Mg]_exch_ + [Zn^2+^] ↔ [Zn]_exch_ + [Ca^2+^ + Mg^2+^](4)
where [Ca + Mg]_exch_ and [Zn]_exch_ are the exchange cations in the composition of CEC, mmol kg^−1^ (determined using AAB-4.8 before the negative reaction to Ca^2+^ ions); [Ca^2+^ + Mg^2+^], and [Zn^2+^] are the same cations in the composition of the soil solution, mM. The proposal to collectively consider Ca^2+^ and Mg^2+^ ions in the soil–soil solution system as cations of the same type, denoting them with the symbol of the predominant Ca cation, was formulated by Beckett [49,50] with reference to the studies of other authors and their own previously obtained results.

Accordingly, the ratio of exchanging cations in the CEC phase and the soil solution can be expressed using Equation (5). The form of this equation shown below reflects the fact that in the case of an equivalent exchange—for example, Zn/(Ca + Mg) − *AR* = *AR*_Zn,(Ca+Mg)_ = [Zn^2+^]/[Ca^2+^ + Mg^2+^]:(5)$$\frac{{\left[\mathrm{Zn}\right]}_{\mathrm{exch},i}}{{\left[\mathrm{Ca}+\mathrm{Mg}\right]}_{\mathrm{exch},i}}=K_{\mathrm{Zn},\left(\mathrm{Ca}+\mathrm{Mg}\right)}^{S}{\frac{{\left[{\mathrm{Zn}}^{2+}\right]}_{i}}{\left[{\mathrm{Ca}}^{2+}+{\mathrm{Mg}}^{2+}\right]}}_{i}$$
where *i* = 1 or 2 is the index characterizing the state of the system, *a*_Ca+Mg_ is the total activity in a quasi-equilibrium soil solution of Ca^2+^ + Mg^2+^ions, and *a*_Zn_ is the activity of Zn^2+^ ions. If we express the concentrations of exchange-bound Zn in the soil for states “1” and “2” through Equation (5), then the change in the amount of the corresponding metal form in the CEC can be described using the following equation: (6)$${\left[\mathrm{Zn}\right]}_{\mathrm{exch},1}-{\left[\mathrm{Zn}\right]}_{\mathrm{exch},2}=K_{\mathrm{Zn},\left(\mathrm{Ca}+\mathrm{Mg}\right)}^{S}\times \left(AR_{2}\times {\left[\mathrm{Ca}+\mathrm{Mg}\right]}_{\mathrm{exch},2}-AR_{1}\times {\left[\mathrm{Ca}+\mathrm{Mg}\right]}_{\mathrm{exch},1}\right)$$

Or, due to the low loading of the CEC with Zn^2+^ ions, it can be expressed as:

[Ca + Mg]_exch,1_ ≈ [Ca + Mg]_exch,2_ = [Ca + Mg]_exch_ obtain:(7)$$\Delta {\left[\mathrm{Zn}\right]}_{\mathrm{exch}}=K_{\mathrm{Zn},\left(\mathrm{Ca}+\mathrm{Mg}\right)}^{S}\times {\left[\mathrm{Ca}+\mathrm{Mg}\right]}_{\mathrm{exch}}\times \Delta AR$$

Or, introducing the designation $PBC_{\mathrm{Zn}}=K_{\mathrm{Zn},\left(\mathrm{Ca}+\mathrm{Mg}\right)}^{S}\times {\left[\mathrm{Ca}+\mathrm{Mg}\right]}_{\mathrm{exch}}\;$, it can be expressed as:(8)$$\Delta {\left[\mathrm{Zn}\right]}_{\mathrm{exch}}=PBC_{\mathrm{Zn}}\times \Delta AR$$

Considering Expressions (5), (7), and (8), the potential buffering capacity of the soil for the linear part of the ion exchange sorption isotherm of Zn can be expressed as the following expressions:(9)$$PBC_{\mathrm{Zn}}=K_{Zn,\left(\mathrm{Ca}+\mathrm{Mg}\right)}^{S}\times {\left[\mathrm{Ca}+\mathrm{Mg}\right]}_{\mathrm{exch}}$$
(10)$$PBC_{\mathrm{Zn}}=\frac{{\left[\mathrm{Zn}\right]}_{\mathrm{exch}}}{AR_{\mathrm{Zn},\left(\mathrm{Ca}+\mathrm{Mg}\right)}}$$
(11)$$PBC_{\mathrm{Zn}}=\frac{{\left[\mathrm{Zn}\right]}_{\mathrm{exch}}}{\left[{\mathrm{Zn}}^{2+}\right]/\left[{\mathrm{Ca}}^{2+}+{\mathrm{Mga}}^{2+}\right]}\;$$
(12)$$PBC_{\mathrm{Zn}}={\left[\mathrm{Ca}+\mathrm{Mg}\right]}_{\mathrm{exch}}\times \frac{K_{d,\;exch}\left(\mathrm{Zn}\right)}{K_{d,\;exch}\left(\mathrm{Ca}+\mathrm{Mg}\right)}$$
where *K_d,exch_*(Zn) = [Zn]_exch_/[Zn]_soil solution_ and *K_d,exch_*(Ca + Mg) = [Ca + Mg]_exch_/[Ca^2+^ + Mg^2+^]_soil solution_, which represent the distribution coefficients of Zn^2+^ and (Ca^2+^ + Mg^2+^), respectively, between the exchange form in the soil (mmol kg^−1^) and quasi-equilibrium soil solution (mM).

It follows that the above conclusions would be correct if the load of CEC with Zn^2+^ ions is higher than 1%. In this case, highly selective CEC sorption sites with respect to zinc ions will be completely blocked, and low-selective sites will remain available for ion exchange sorption, [51]. At the same time, the CEC loading of the studied cation should not exceed 5–10% (according to F. Beckett [50]). If the above conditions are met, the following would be observed: *K*^S^_Zn,(Ca+Mg)_ = const, where *K*^S^_Zn,(Ca+Mg)_ is the selectivity coefficient of ion exchange Zn^2+^/(Ca^2+^ + Mg^2+^) [52,53]. The specified range was determined using the ion exchange equilibrium method and amounted to 0.5–19.6% of the total CEC of the studied soil. The average value of *K^S^*_Zn,(Ca+Mg)_ was equal to 13.5 ± 6.6. It should also be noted that the range of non-toxic concentrations of zinc in the soil fully fit into this range of CEC loadings, which corresponded to the indicative uptake of Zn by barley.

Considering that in the range of non-toxic concentrations of Zn in the soil and, accordingly, the soil solution, the values of *CR*_Zn,exch_ = [Zn]_plant_/[Zn]_exch_ and *CF_Zn_ =* [Zn]_plant_/[Zn]_soil solution_ were found to be constant, so we substituted them in the *K_d,exch_*(Zn) parameter in Equation (12).

We also considered that for VPs (indicator part of barley plants):

- *CR*_(Ca+Mg),exch_ = *const* (because of the constant concentration of Ca and Mg in the soils, which have been applied with the increasing doses of Zn, i.e., [Ca + Mg]_exch_ = *const* = 23.7 ± 4.4 mmol kg^−1^).

- *CF*_(Ca+Mg)_ = *const* (because the concentrations of Ca^2+^ and Mg^2+^ in the soil solution with an increasing concentration of Zn^2+^ was not changed, and the [Ca + Mg]_plant_/[Ca^2+^ + Mg^2+^]_soil solution_ ratio remained constant in the experiment with the water culture of barley. It was found to be equal to 266 ± 91 dm^3^ kg^−1^.

Based on these arguments, we express the parameters *K_d,exch_*(Zn) and *K_d,exch_*(Ca + Mg) through the parameters *CR*_Zn,exch_, *CR*_(Ca+Mg),exch_, *CF_Zn_*, and *CF*_(Ca+Mg)_ in Equation (12).

We obtained the final form of the equation for calculating the potential buffer capacity of soils with respect to technogenic Zn:(13)$$PBC{\left(V\right)}_{\mathrm{Zn}}={\left[\mathrm{Ca}+\mathrm{Mg}\right]}_{\mathrm{exch}}\times \frac{CF_{\mathrm{Zn}}\times CR_{\left(\mathrm{Ca}+\mathrm{Mg}\right),\mathrm{exch}}}{CF_{\left(\mathrm{Ca}+\mathrm{Mg}\right)}\times CR_{\mathrm{Zn},\mathrm{exch}}}$$

With the help of this equation while knowing the parameters *CF*_Zn_ and *CF*_(Ca+Mg)_ for a specific agricultural crop (they can be obtained from experiments with water crops), it is easy to calculate the values of the potential buffer capacity of any soil with respect to zinc, for example, from agroecological examination materials in agricultural lands where amounts of Ca, Mg, and Zn in soil (exchangeable form) and conjugate plant samples have been determined.

Thus, considering the sum of green leaves and stems as indicator part of barley plants, we obtained the following results of experiments I and II (Table 2).

After substituting the values of the corresponding parameters into Equation (13), we obtained the value of *PBC(V)_Zn_* for the studied Albic Retisol (Loamic, Ochric) soil:*PBC(V)*_Zn_ = [Ca + Mg]_exch_×(*CF*_Zn_ × *CR*_(Ca+Mg), exch_)/(*CR*_Zn,exch_ × *CF*_(Ca+Mg)_) = 23.7 × (640 × 22.2)/(5.68 × 339) = 175 mmol kg^−1^(14)

Thus, using *CF*_Me(VPs)_, (Me = Ca + Mg, Zn) with values established in experiments with aquatic crops of agricultural plants, it is possible to quantify the buffering capacity of soils based on data on the content of a “labile (accessible to plants)” forms of Zn in these soils and the metal concentrations in the vegetative mass of plants.

In parallel, the *PBC*_Zn_ of the studied Albic Retisol (Loamic, Ochric) soil was determined with the classical ion exchange equilibrium method under static conditions using a modification of the Beckett method [47,48,49,50]. In this method, the concentration of the macronutrient cation—Ca^2+^ in the balancing solution, with an increase in the amount of the dissolved metal under study (Zn in our case)—remained unchanged and equal to 2 mM (20 mM in the Beckett method), which is close to the concentration of the macronutrient cation in the free (gravity) soil solution of the humus horizons of most soils. The value of *PBC*_Zn_ was determined to be 150 mmol kg^−1^. This value was comparable to the *PBC(V)*_Zn_ value, obtained using test plants, despite the fundamental difference between the two methods (the relative difference was 15%).

### 2.4. Experiment III

During the vegetation experiment with a flow lysimeter, it was found that the decrease in the concentration of stable Zn and the volumetric activity density of ^65^Zn in the soil solution before and after the vegetation vessels over time was satisfactorily described by a power function with a negative indicator of the type [Zn]_soil solution_ = *a*
*×*
*t^b^*, where ***a*** and ***b*** are parameters and ***t*** is time in days (Figure 6a,b).

The values of parameters ***a*** and ***b*** and the determination coefficient (R^2^) for Zn(^65^Zn) in the soil solution at the output of the lysimeter (before the vegetation vessels) were: 263, −0.81 (0.85) and 666, −0.71 (0.90), respectively. In the soil solution after leaving the vegetation vessels and coarse filtration, they were: 262, −0.89 (0.90) and 609, −0.76 (0.90), respectively.

During the vegetative experiment, a significant decrease in the concentration of [NO_3_^−^] (mg dm^−3^) ions in the lysimetric solution was observed, while the average (at the input and output from the vegetative vessels) pH value increased from 5 to 7 (Figure 7a,b). The concentrations of [K^+^] and [NH_4_^+^] ions in the solution also decreased during the vegetation experiment, respectively, from 40 ± 6 to 11 ± 0.5 mg dm^−3^ and from 6.0 ± 0.9 to 0.6 ± 0.1 mg dm^−3^.

The values of the specific activity of ^65^Zn/Zn in studied objects are important indicators for assessing the contribution of a particular form of zinc to the soil–soil solution–plant migration chain.

Data on the dynamics of the specific activity of ^65^Zn/Zn in the soil solution, vegetative parts (VPs), and roots of the test plant (barley) in terms of stable Zn contained in this solution (in Bq mg^−1^) are shown in Figure 8a–c. The average values for the vegetation period of the corresponding parameters A_sp_(^65^Zn/Zn)_soil solution_, A_sp_(^65^Zn/Zn)_VPs_, A_sp_(^65^Zn/Zn)_roots_ were, respectively, 3580 ± 390, 3210 ± 1250, and 3230 ± 780 Bq mg^−1^. As a result of the alkalinization of the lysimetric solution, its extracting ability with respect to the “freshly applied” ^65^Zn gradually decreased, although not as much as with respect to the less labile native zinc. This led to pronounced trends in increasing values of A_sp_(^65^Zn/Zn)_soil solution_, A_sp_(^65^Zn/Zn)_VPs_, and A_sp_(^65^Zn/Zn)_roots_ during the growing season.

The data obtained on the dynamics of the content of Zn(^65^Zn) in the vegetative parts of barley (data for roots are not given) showed that the content of zinc in the plants generally increased during ontogenesis (Figure 9a,b).

The values of the Zn(^65^Zn) concentration factors grew with increasing plant age (Figure 10) due to both the accumulative effect when zinc is absorbed by plants and decreases in its content in the soil solution during the experiment. Due to the high variability of the data, it was not possible to identify a significant difference in the values of *CF*_Zn_ and *CF*_Zn-65_.

In order to study the contribution of different soil forms to Zn content of the liquid phase of the soil in more detail, we used ^65^Zn as a radioactive tracer when applying a sequential fractionation scheme of selected soil samples in accordance with the modified BCR method. The obtained results are presented in Figure 11a–c. A comparative analysis of the data showed the following ratios of different forms of stable Zn and radionuclide ^65^Zn (data shown in parentheses) in soil in percent: I. 34.2 ± 3.4 (10.7 ± 0.4); II. 31.9 ± 3.7 (17.3 ± 0.1); III. 9.9 ± 2.6 (27.8 ± 2.3); and IV. 24.0 ± 7.7 (42.6 ± 3.9). According to Figure 11a,b, the relative contents of labile and conditionally labile forms of ^65^Zn in the soil (Fractions I and II) significantly exceeded the content of the corresponding forms of the stable (natural) isotope Zn, respectively, by 3.2 and 1.8 times.

At the same time, the values of the relative content of conditionally fixed and fixed forms of ^65^Zn in the soil (fractions III–IV) were significantly lower than that of Zn by 2.8 and 3.0 times, respectively.

The *Enrichment Factor* values of the corresponding forms of natural Zn with the radioactive tracer ^65^Zn in relation to the main component of our model system—the lysimetric soil solution at the time corresponding to the beginning of the growing experiment—were: 1.54 ± 0.11, 0.82 ± 0.07, 0.19 ± 0.03, and 0.24 ± 0.03 (Figure 11c). The value of *A*_sp_(^65^Zn/Zn)_soil solution_ was equal to 3220 ± 110 Bq mg^−1^. The ordinate 1.0 secant value was an *EF* (^65^Zn/Zn)_soil solution_.

The cumulative form of *EF* data obtained are represented as a curve (Figure 11d) on which each subsequent *EF* value is an integral value, where the ratio of the total specific activity of ^65^Zn of this and previous fractions (Bq kg^−1^) to the total mass fraction of the corresponding fractions of stable Zn (mg kg^−1^ of a soil) is normalized to the specific activity of ^65^Zn/Zn in soil solution: *EF**_Σ_*_Fr.#_(^65^Zn/Zn) = A_sp_(^65^Zn/Zn)*_Σ_*_Fr.#_/A_sp_(^65^Zn/Zn)_soil solution_. The values of *EF* for the sum of the relevant forms of natural Zn with tracer ^65^Zn in relation to the soil solution were 1.54 ± 0.11 (Fraction I), 1.08 ± 0.03 (Σ of Fractions I–II), 0.64 ± 0.02 (Σ of Fractions I–III), and 0.48 ± 0.03 (Σ of Fractions I–IV). The secant value of the coordinate 1.00 was the *EF* (^65^Zn/Zn)_soil solution_.

### 2.5. Mobility of Native (Natural) Zn and Technogenic Zn in the Soil Solution–Plant System (According to Lysimetric Experience)

For stable Zn and radionuclide ^65^Zn, it was found that their concentration and volumetric activity density in the soil solution decreased over time (Figure 6a,b), and at the input of vegetative vessels, the corresponding concentrations were higher than at the output from the vegetative vessels, which can be explained by the uptake of zinc by barley roots. The decrease in the concentration of stable Zn and the volumetric activity density of radionuclide ^65^Zn in soil solutions over time were associated with both the depletion of the pool of water-soluble forms of metals in the soil as a result of continuous uptake by plant roots (not compensated for by desorption of Zn (^65^Zn) from the cation exchange complex (CEC) into the soil solution) and the alkalinization of the soil solution by root secretions.

The decrease in the concentration of nitrate, as well as ammonium and potassium ions, in the soil solution (Figure 7a) can be described by exponential equations. It should be noted that the high concentration of nitrates led to an increase in the alkalinity of the quasi-equilibrium soil solution during the growing season (Figure 7b) due to the ability of barley to alkalize nutrient solutions in the light, releasing HCO_3_^−^ or OH^−^ ions in the presence of a sufficient amount of nitrate ions and a low content of ammonium ions. This fact has been reported by many researchers [7,40,54]. It is caused by compliance with the principle of electroneutrality during the transmembrane transfer of anions and cations to the root symplast.

It was found that the content of Zn (^65^Zn) in plants increased during ontogenesis (Figure 9a,b). This fact indicates that zinc is continuously accumulated in plants during the growth and development phases (before maturation) during ontogenesis, and this should be considered when comparing data on the metal contents in the vegetative mass of plants selected at different stages of development. We attribute a very significant increase in the values of *CF*_Zn_ and *CF*_Zn-65_ during ontogenesis to both the accumulative effect of zinc plants uptake and a decrease in its content in the soil solution during the experiment (Figure 10).

### 2.6. Assessment of the Mobility of Natural Zn in the Soil–Soil Solution–Plant System Using the Radioisotope ^65^Zn as a Tracer

We assumed that the isotope ^65^Zn introduced into the soil as a radioactive label was not evenly distributed between the various forms of natural (stable) Zn, but it turned out to be primarily bound in the form of labile forms. In the future, its transformation into less labile forms should theoretically take place. This process is very long, so much so that it will not be possible to trace it for ^65^Zn (during this time, the radionuclide will repeatedly decay). Nevertheless, it is possible to consider some theoretical aspects of the problem related to the direction and speed of transformation of ^65^Zn forms in the soil.

It is known that in the case of the isotopic exchange of ^65^Zn/Zn, we are dealing with an “ideal isotopic exchange” in which isotopic atoms that are identical in their physicochemical properties participate. In our case:Zn(soil form) + ^65^Zn^2+^(soil solution) ↔ ^65^Zn(soil form) + Zn^2+^ (soil solution)(15)

The processes of ideal isotope exchange are characterized by the absence of elemental (chemical) changes, as well as the immutability of the number of interacting particles and their concentrations. The reason for the spontaneous flow of ideal isotope exchange processes is only an increase in the entropy of the system, since the change in its enthalpy in this case will be zero (Δ*H* = 0). From a physical point of view, the increase in the entropy of the system during isotope exchange corresponds to the transition of the system from a more ordered state (different amounts of the isotope ^65^Zn are present in different forms) to a less ordered one (the isotope is evenly distributed between the forms involved in the isotope exchange process), which corresponds to the mixing of isotopes. When an equilibrium state occurs with an ideal isotope exchange, Δ*G*^0^ = 0. Accordingly, the equilibrium constant (*K_p_*) = 1 [29]. For the above equation of the ideal isotope exchange ^65^Zn/Zn, this ratio looks like:*K_p_* = ([^65^Zn]_soil_ × [Zn]_soil solution_)/([Zn]_soil_ × [^65^Zn] _soil solution_) = 1(16)

This leads to the identity of the isotopic composition of the exchanging forms:([^65^Zn]/[Zn]) _soil_ = ([^65^Zn]/[Zn]) _soil solution_(17)

In the case of a non-equilibrium state (as in our case), we present the following inequality:([^65^Zn]/[Zn]) _soil_ ≠ ([^65^Zn]/[Zn]) _soil solution_(18)

The value of ([^65^Zn]/[Zn])_soil_/([^65^Zn]/[Zn])_soil solution_ is the same as *EF*(^65^Zn/Zn)_soil_. The higher the value of *EF* (^65^Zn/Zn)_soil_, the further away the soil is from the state of isotopic equilibrium between radioactive and naturally stable zinc isotopes present in the solid phase in all forms (labile, conditionally labile, and fixed) on the one hand and the water-soluble form on the other hand. Due to the insignificance of the values of [^65^Zn]_soil_ and [^65^Zn] _soil solution_, they are usually expressed by the values of mass and volumetric activity density, respectively, of the radionuclide: A_m_(^65^Zn)_soil_, A_v_(^65^Zn)_soil solution_.

The solution is the most important effective phase of soil [52] at the boundary of which with solid phases ion exchange processes occur (transformation function) and a certain quasi-equilibrium state is established, as described, for example, by Equation (15). However, due to its special properties, such as its high mobility (fluidity) and high rate of diffusion transfer of dissolved substances, it acts as a connecting link due to which ion-exchange reactions (for example, the zinc isotopes considered in this paper) indirectly occur between various forms localized in different parts of the solid phase of the soil (transport function). As a result, the total net process look such that the ^65^Zn present in some forms in the solid phase will be predominantly desorbed into the soil solution, passing into other forms.

If A_sp_(^65^Zn/Zn)_Fr.#_ > A_sp_(^65^Zn/Zn)_soil solution_, i.e., the enrichment factor of any form of Zn in the soil with radionuclide ^65^Zn relative to the soil solution is greater than 1, then the process of isotopic exchange of radionuclide between the corresponding form and the solution is shifted towards the latter. The place of ^65^Zn^2+^ ions in the soil is occupied by ions of the natural stable isotopic carrier Zn^2+^ from soil solution. If A_sp_(^65^Zn/Zn)_Fr.#_ < A_sp_(^65^Zn/Zn)_soil solution_, the reverse process occurs. Thus, through the liquid phase, as already noted, there is an exchange of Zn(^65^Zn) between competing binding sites in the solid phase of soils, forming with zinc ions the corresponding chemical forms. The concentration of [Zn] and the volumetric activity density of ^65^Zn in a quasi-equilibrium soil (lysimetric) solution reflect the contribution of it different forms in the studied soil.

Values of the enrichment factor of the labile chemical Zn fraction (Fraction I) via the radioactive tracer ^65^Zn (*EF* (^65^Zn/Zn)_Fr.I_ = 1.54 > 1, while *EF* (^65^Zn/Zn)_Fr.II_ = 0.82, *EF* (^65^Zn/Zn)_Fr.III_ = 0.19, and *EF* (^65^Zn/Zn)_Fr.IV_ = 0.24 were less then 1 (Figure 11c). This allowed us to assume that in the soil–soil solution system from chemical Fraction I, ^65^Zn^2+^ ions were predominantly desorbed into the soil solution, and for fractions II–IV, in contrast, were sorbed from the solution, which gradually led to a decrease in A_sp_(^65^Zn/Zn) for the first fraction and an increase for the second group of fractions.

Based on the obtained results and the assumption of the increasing ability of extractants used in the sequential extraction of zinc ions [20], it was possible to calculate the pool of labile zinc in the unit of soil mass—the “*E*-value”: (*E*_Zn_) = C(Zn)_Fr.I_ = 3.95 ± 0.16 mg kg^−1^ (or 10.7 ± 0.4%). The corresponding specific activity value was ^65^Zn (A_sp_(^65^Zn/Zn)_Fr.I_) equal to 4660 ± 830 Bq×mg^−1^.

Attention is drawn to the fact that the average value during the growing season A_sp_(^65^Zn/Zn)_soil solution_ = 3580 ± 390 Bq mg^−1^ < A_sp_(^65^Zn/Zn)_Fr.I_. This allowed us to assume the existence of a slow process of isotopic exchange between mobile and other (“*conditionally labile*”, “*conditionally fixed*”, and “*fixed*”) forms of zinc by means of the liquid phase of the studied soil–soil solution. It was possible to fix this moment thanks to a long process of preliminary equilibration of ^65^Zn with the studied soil.

Later, during the vegetation experiment, as a result of the vital activity of plants, the soil solution was depleted with zinc and alkalized, which led to an increase in the value of A_sp_(^65^Zn/Zn)_soil solution_ to values close to A_sp_(^65^Zn/Zn)_Fr.I_ (Figure 9a), which actually reflected the ratio of ^65^Zn/Zn in labile and “*conditionally labile*” forms in the soil and their contribution to the composition of a quasi-equilibrium soil solution.

For balance calculations and the study of the transformation of the forms of Zn(^65^Zn) in the soil, data on the total removal of both natural and radioactive zinc isotopes from the soil–soil solution system by vegetative parts and barley roots are of particular interest (recall that five plant selections were made in total). We found that the percentage of the total amount of Zn(^65^Zn) contained in the soil was insignificant and amounted to only 0.34 (0.70)%. Consequently, the removal of metal by plants had no noticeable effects on the ratio of the forms of zinc in the soil.

## 3. Materials and Methods

The behavior of Zn in a soil–plant system was studied in vegetation experiments with soil culture with increasing zinc concentration in the soil (greenhouse conditions), in vegetation experiments with water culture with increasing zinc concentration in a soil nutrient solution (greenhouse conditions), and in a lysimetric experiment with an aqueous culture with the sole application of radioactive ^65^Zn to the soil (laboratory conditions).

Barley (*Hordeum vulgare* L.) of the Zazersky-85 variety and soddy-podzolic sandy loam cultivated soil—Albic Retisol (Loamic, Ochric) not containing free carbonates selected from the arable horizon of agricultural land near Obninsk (Kaluga region, Russia) were selected as our objects of research.

### 3.1. Experiment with the Soil Culture of Barley

During the research, most attention was paid to the root uptake and redistribution of Zn between different parts of barley plants. To do this, 5 kg of air-dry Albic Retisol (Loamic, Ochric) soil were placed into vessels. Nutrients were applied to the soil in the form of aqueous solutions of KH_2_PO_4_ and K_2_SO_4_ salts at rates of 100 mg kg^−1^ P and K, respectively, and Zn was applied in the form of nitrate solutions.

The amount of Zn applied to the soil was as follows: 0, 25, 50, 100, 175, 250 and 500 mg kg^−1^. The amount of nitrogen introduced with Zn(NO_3_)_2_ was adjusted according to the variant with the sub-maximum dose of Zn using NH_4_NO_3_ (but no more than 1 g N/vessel). After applying salt solutions, the soils in the vessels were incubated for 30 days at a temperature of 20–23 °C. Twenty five barley seeds were sown in each vessel. The plants on the green mass were harvested at 7, 14, 21, 30, 45 and 70 days after sowing. The experiment was carried out in triplicate.

The experiment was carried out at a temperature of 20–29 °C, relative humidity of 55–75%, and soil mass moisture content of 60% of the full water capacity (FWC).

Exchangeable potassium, calcium, and magnesium were extracted from soils using different reagents: 1 M NH_4_Cl (pH 6.5); 1 M CH_3_COONH_4_, pH 7.0 (AAB-7.0); and 1 M CH_3_COONH_4_, pH 4.8 (AAB-4.8) [55], respectively, before a negative reaction to Ca^2+^ ions. The physical and chemical parameters of the studied soil were determined with conventional methods [55,56]: pH_KCl_ (pH_water_) was determined with a potentiometric method in a suspension of soil in 1 M solution of KCl (distilled water) with a ratio of solid and liquid phases (S:L) = 1:2.5 (1:25 for peat soil), the granulometric composition of soil was determined with the pipette method of N.A. Kachinsky [48,55], humus content was determined by means of bichromatic oxidation by the Tyurin method, hydrolytic acidity was determined by means of the Kappen method, the sum of exchange bases was determined by means of the Kappen–Gilkovitz method, and the content of labile forms of P_2_O_5_ was determined by means of by the Kirsanov method in the modification of TSINAO (0.2 M HCl, S:L = 1:5).

The combination of easily and difficult-to-exchange forms of Zn(^65^Zn) [18,34,57] was extracted using 1 M CH_3_COONH_4_ at pH 4.8 with the modified method proposed by N.G. Zyrin [32]—by successive, exhaustive extractions to a negative reaction to Ca^2+^. The potential buffering capacity of soils with respect to Zn was determined by the Beckett method [47,49,50] using 0.01 M CaCl_2_.

### 3.2. Experiment with Water Culture of Barley

Our experiments were carried out on constantly aerated and mixed nutrient solutions extracted from the studied soil at a narrow soil:solution ratio = 1:2, in which, in addition to the background concentration (0.07 µM), Zn was added in the form of nitrate at the following concentrations: 0,0.63, 5.8, 29, 57, 114 and 430 µM. In the soil solutions, the amount of applied Zn(NO_3_^−^)_2_ of N(NO_3_^−^) was adjusted according to the variant with a 114 µM dose of HM using Ca(NO_3_)_2_ (“puriss”) at the rate of 0.2 mM N(NO_3_^−^). The contents of macroelement cations in nutrient solutions were (in mM): 1.60 ± 0.05 Ca^2+^ (considering the added in the form of Ca(NO_3_)_2_); 0.16 ± 0.02 Mg^2+^, and 0.38 ± 0.02 K^+^. The pH value of the nutrient solutions was adjusted to 5.8, corresponding to the acidity of a solution containing 114 µM Zn^2+^, using solutions of CitH_3_ and NaOH. The total content of citric acid anion in nutrient solutions was 0.05 mM. The plants were grown for 21 days, and we changed the solution to a fresh one daily.

### 3.3. Lysimetric Experiment with Water Culture of Barley

The vegetation experiment was carried out using a special stand, including a flow lysimetric installation of an original design (Figure 12 and Figure A1 in Appendix A). In addition to the cyclic lysimetric installation, which provided the gravity runoff of soil moisture, the stand included:

- Flowing vegetative vessels with a soil solution in which sprouted plant seeds were placed on special stands in containers with a mesh bottom (nylon fabric) filled with coarse sand.

- A peristaltic pump.

- A system of tubes, taps, and adapter tees; a buffer tank.

- A lighting source for plants. From above, the flow lysimeter was non-hermetically covered with a plexiglass lid with a sprinkler device.

The preliminary preparation of the soil included the introduction of 200 kBq kg^−1^ of radionuclide ^65^Zn (T_1/2_ = 224 days) in the form of a working solution of ^65^Zn(II) without an isotopic carrier such as stable Zn. The working solution was prepared from a sample solution of ^65^Zn(II) in 0.1 M HCl (“CYCLOTRON Co., Ltd.”, Obninsk, Russia), which contained 42.4 MBq ^65^Zn at the time of certification. The mass activity density of the soil due to the presence of the radioisotope ^65^Zn (Am^65^Zn_soil_) at the beginning of the vegetation experiment under consideration was 68,700 ± 2800 Bq kg^−1^.

The resulting soil suspension after the application of the radionuclide was thoroughly mixed and incubated at room temperature for ½ year, moistened twice a month, and dried in air (preventing complete drying). Then, the dried soil was ground and passed through a sieve with a diameter of 2 mm. The soil prepared in this way, containing ^65^Zn, was placed in layers in a lysimetric installation with alternating layers of soil and washed quartz sand with a particle diameter of <1 mm as drainage (Figure 12). To prevent the colmatation of soil pores under the gravitational current of moisture, soil/sand layers were vertically oriented.

The total amount of deposited sand and soil was 5 kg each. The thickness of the layers was 2.5 cm. After that, the soil–soil solution system has been balanced in the lysimeter for 5 months by pouring deionized water onto the soil surface in the lysimeter and periodically returning the water flowing from the lysimeter back to the soil surface. Three weeks before the start of the vegetation experiment, a growing installation was assembled and water was added to the system up to a total volume of 5.5 dm^3^ along with the nutrient solutions at the rates of 100 mg kg^−1^ N and K in the forms of NH_4_NO_3_ and K_2_SO_4_, respectively. Additional phosphorus was not applied, since the content of its labile forms in the soil was sufficient (Table 1). After assembly, the lysimetric installation and vegetative vessels were wrapped with a light-tight film. Then, the peristaltic pump was started and the system was left in operation for balancing for 3 weeks. Considering that the total volume of the liquid phase in the system was about 5.5. dm^3^ and the rate of water supply by the peristaltic pump through the sprinkler device was 4 cm^3^ per minute, the soil solution was subjected to complete regeneration, passing through the lysimeter, during the day.

The plants were grown in glass vegetation vessels with a volume of 2 dm^3^ in triplicate. The volume of the soil lysimetric solution in each vessel was approximately 1.25 dm^3^. The mixing of solutions was carried out by continuously bubbling air supplied to each vegetative vessel through thin silicone tubes using a low-power compressor. To compensate for evaporating moisture, the total volume of deionized water added daily to the soil surface in the lysimeter was 150–200 mL. In each vegetative vessel, a plastic separator was placed on stands with 6 large holes into which cartridges filled with large washed quartz sand (2–3 mm), closed from below with a nylon cloth, were inserted. Three barley seedlings sprouted within 3 days were planted in each cartridge. When barley roots appeared outside the cartridges, the latter were slightly raised above the water surface, ensuring that the roots were in the water. At the same time, the movement of water in the vegetative vessels caused by bubbling was sufficient to wet the substrate (sand) inside the cartridges. The duration of the vegetation experiment was 35 days. In total, 5 plant selections were made during the experiment each week (the 1st selection was double due to the small amount of plant material).

### 3.4. Determination of Zn(^65^Zn) Forms (Chemical Fractions) in the Soil

Using AAB-4.8, “labile (available to plants)” forms of zinc were extracted from the soil using successive exhaustive extractions before a negative reaction to Ca^2+^ ions [32,35].

To determine the content of ^65^Zn(Zn) associated with different organo–mineral fractions with the BCR method [21] (Table 3), the soil samples were preliminarily prepared.

Various forms of ^65^Zn (Zn) were determined in the soil located in the lysimeter at moisture content (W = FMC (field moisture capacity)) just before the vegetation experiment. A mixed soil sample was obtained by combining micro-samples (m ≈ 5 g) extracted with the use of a special bore made of a chemically inert material (a total of 10 injections). After the careful homogenization of the combined sample, subsamples of raw soil material with known humidity underwent successive chemical extraction in accordance with the BCR fractionation scheme.

In order to facilitate the further analysis and discussion of the obtained data, we introduced the special classification of the forms of Zn(^65^Zn) in the soil: exchange and carbonate bonds denoted as “*labile*”; bonds associated with recoverable Fe–Mn oxides denoted as “*conditionally labile*”; bonds associated with oxidizable organic matter and sulfides denoted as “*conditionally fixed*”; and residue denoted as “*fixed*”.

### 3.5. Elemental Analysis and γ-Spectrometry of Samples

During the vegetation experiments, the following dynamics were determined:

- Concentrations of zinc and the mass activity density of ^65^Zn in soil (including different forms) and plants: [Zn]_soil_ and [Zn]_plant_ (mg kg^−1^); A_m_(^65^Zn)_soil_ and A_m_(^65^Zn)_plant_, (Bq kg^−1^).

- Concentration of Zn and the volumetric activity density of ^65^Zn in the phase of the soil solution: [Zn]_soil solution_ (µg L^−1^) and A_v_(^65^Zn)_soil solution_ (Bq L^−1^), respectively.

Values of mass and volumetric activity density of ^65^Zn were calculated on the date of the beginning of the vegetation experiment.

Based on the above parameters, the values of the key parameter, conventionally called the “specific activity of ^65^Zn/Zn”, were calculated in lysimetric waters, individual chemical fractions (forms) of zinc in the soil, and parts of plants. These were determined as the ratio of mass or volumetric activity density to the concentration of stable Zn contained in the corresponding objects of study (Bq mg^−1^):
- In solutionA_sp_(^65^Zn/Zn)_solution_ = A_v_(^65^Zn)/[Zn]_solution_- In the soil as a whole, and in individual chemical fractions of the soilA_sp_(^65^Zn/Zn)_Fr.#_ = A_m_(^65^Zn)_Fr.#_/[Zn]_Fr.#_- In vegetative parts (VPs) of plantsA_sp_(^65^Zn/Zn)_VP_ = A_m_(^65^Zn)_VP_/[Zn]_VP_- In rootsA_sp_(^65^Zn/Zn)_roots_ = A_m_(^65^Zn)_roots_/[Zn]_roots_

Using the specific activity of ^65^Zn/Zn, the values of the parameter we called *“*Enrichment Factors—*EF*(^65^Zn/Zn) (or simply *EF*)—were calculated. These factors comprised the enrichment with the radioactive tracer ^65^Zn of natural (stable) Zn contained in soil (including individual chemical fractions) and plants in relation to the main component of our model system–soil solution, e.g., *EF*_Fr.#_(^65^Zn/Zn) = A_sp_(^65^Zn/Zn) _Fr.#_/A_sp_(^65^Zn/Zn)_soil solution_.

The concentrations of Zn (^65^Zn) in the roots and vegetative parts (VP) of barley plants were determined after the dry ashing of samples at 500 °C followed by the acid extraction of metal HNO_3_; the total (gross) mass fraction of elements in the soil was determined by the Obukhov–Plekhanova method [58] after ashing at 500 °C and the subsequent decomposition of samples using heating with an HCl_conc._+HNO_3conc._+HF_conc._ mixture followed by successive treatments by HF_conc_ (twice) and HNO_3conc_ (triplicate).

Elemental analysis was carried out via the atomic absorption and optical emission methods with inductively coupled plasma using 140AA (Agilent) and Liberty II (Varian) spectrometers. The mass and volumetric activity densities of ^65^Zn were determined on a gamma-spectrometric complex consisting of an ORTEC HPGe detector and analyzer (with a relative recording efficiency of 40%) and LSRM SpectraLine GP software.

### 3.6. Statistical Analysis

The accuracy of the approximating equations used to describe the obtained dependencies was estimated using the *t*-criterion and the coefficients of determination R^2^. The statistical analysis of experimental data was carried out by standard methods using MS-Excel based on the theoretical aspects set out in the works [59,60,61].

## 4. Conclusions

For the studied soils, various types of plant responses to changes in the concentration of Zn were determined using barley as the test plant. Ranges of corresponding concentrations in soils and various parts of test plants were established.

It was shown that the edaphic factors determining the buffering capacity of soils play no less important roles in regulating the mobility of Zn in the soil–plant system than the biological factor (physiological characteristics of plants). For Zn, the concentration ratios (*CRs*) and concentration factors (*CFs*) in various parts of plants were found to be constant in the area of the indicative type of plant response to changes in the metal contents in soils (for example, we found values of *CR_labile_* = 5.68 and *CF* = 640 L kg^−1^ for the vegetative mass of barley).

A method was proposed for determining the buffer capacity of soils with respect to HMs (*PBC(V)*_HM_) using barley as the test plant (the tested part of the plant was leaves). With this method, we determined the value of *PBC(V)*_Zn_ in the field of non-toxic zinc concentrations for the studied Albic Retisol (Loamic, Ochric)soil to be 175 mmol kg^−1^. In parallel, the *PBC*_Zn_ of the studied soil was determined by the laboratory method of ion exchange equilibrium under static conditions. The *PBC*_Zn_ value was 150 mmol kg^−1^. Thus, the values of the buffer capacity of soils in relation to a heavy metal (Zn), determined by two methods, turned out to be of the same order. The relative difference was 15%.

The considered methodological approach opened up opportunities for using data obtained during the agroecological monitoring of contaminated agricultural lands (such as the content of labile forms of HMs, the exchange forms of macroelements (Ca and Mg) in soils, and the concentrations of HMs, Ca, and Mg in plants) to calculate the values of *PBC(V)*_HM_ of the studied soils. However, to do this, it is necessary to have a database of the concentration factors of these HMs and macroelements (such as Ca and Mg) for various crops that are intended to be used as test plants. It is desirable that the composition of nutrient solutions be as close as possible to real soil solutions, at least at the level of soil types.

With the help of an original lysimetric installation of cyclic action and the use of a radioactive tracer ^65^Zn applied to the studied Albic Retisol (Loamic, Ochric)soil, various aspects of the process of zinc migration and transformation of its forms in the soil–soil solution–plant system were studied. This allowed us to obtain a number of important parameters characterizing the lability and bioavailability of zinc in quasi-equilibrium conditions. Thus, for the studied soil with a known content of natural (stable) Zn and additionally introduced radionuclide ^65^Zn, the regularities of metal distribution between different soil forms were established by using sequential fractionation method.

It was established that the enrichment of labile chemical Zn fraction by the radioactive tracer ^65^Zn (*EF* (^65^Zn/Zn)_Fr.I_ was 1.54 times higher than the same value for the soil solution at the time corresponding to the beginning of the vegetation experiment. At the same time, the ratio of the remaining (“conditionally labile”, “conditionally fixed”, and “fixed”) forms of stable Zn in the soil with ^65^Zn was 0.19–0.82 times lower than that of the soil solution. Based on the obtained value of *EF* (^65^Zn/Zn)_Fr.I_, we calculated the amount of the pool of labile “*E*-value” (*E*_Zn_) compounds of native zinc in the unit of the mass of the studied soil as: *E*_Zn_ ≅ C(Zn)_, Fr.I_ = 3.95 ± 0.16 mg kg^−1^ (or 10.7 ± 0.4%). The corresponding specific activity value of ^65^Zn (A_sp_(^65^Zn/Zn)_Fr.I_) equaled 4660 ± 830 Bq mg^−1^.

At the beginning of the vegetation experiment, we recorded a lower enrichment of native zinc contained in the quasi-equilibrium soil solution with ^65^Zn (after balancing it with the soil for 1 year) compared to the stable zinc contained in Fraction I. This indicated the existence of a slow process of isotopic exchange between the mobile and other *(*“conditionally labile”, “conditionally fixed”, and “fixed”) forms of zinc by means of the liquid phase of the studied soil–soil solution. Later, during the vegetation experiment, as a result of the vital activity of the plants, the soil solution was depleted with zinc and alkalinized, which led to an increase in the values of A_sp_(^65^Zn/Zn)_soil solution_ to those close to A_sp_(^65^Zn/Zn) _Fr.I_.

The dynamics of the transition to the soil solution phase and the parameters of the uptake of Zn(^65^Zn) by test plants were also evaluated. According to the results of the experiment, the total removal of natural Zn and radionuclide ^65^Zn by plants (roots and vegetative parts) was calculated as a percentage of the total amount of Zn(^65^Zn) in the soil: 0.34 (0.70)%. These data suggest that the removal of zinc from soil by plants is insignificant and had no noticeable effects on the ratio of the forms of the metal in the soil.

## Figures and Tables

**Figure 1 plants-10-02496-f001:**
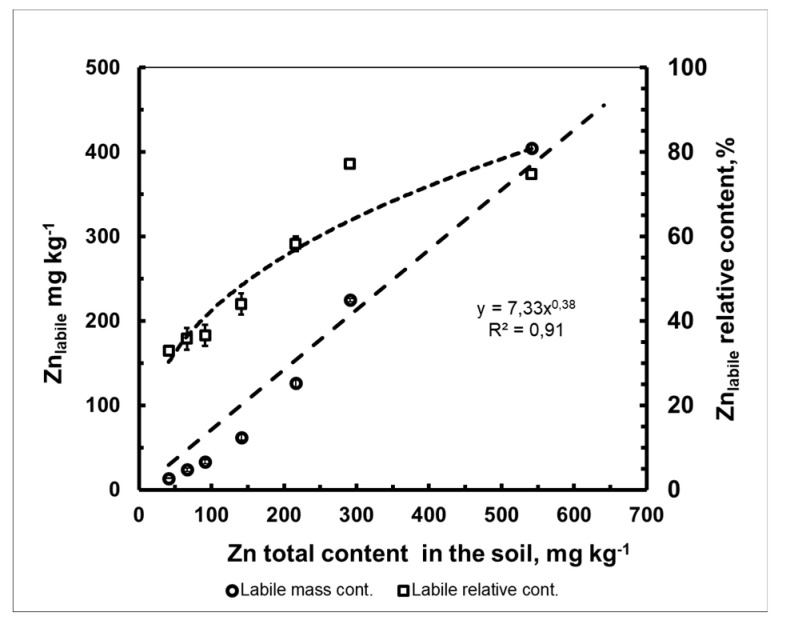
The change in the content of the labile form of Zn in Albic Retisol (Loamic, Ochric) soil (mean value ± standard deviation; *n* = 3). Data are from the vegetation experiment I.

**Figure 2 plants-10-02496-f002:**
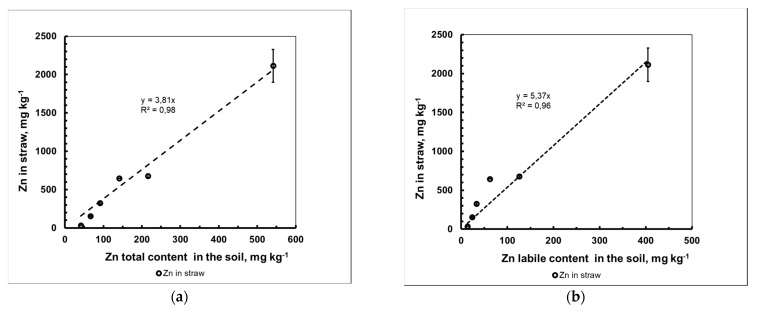
Dependence of the mass concentration of Zn in barley straw on the total amount (**a**) and labile form (**b**) of the metal in the soil (mean value ± standard deviation; *n* = 3).

**Figure 3 plants-10-02496-f003:**
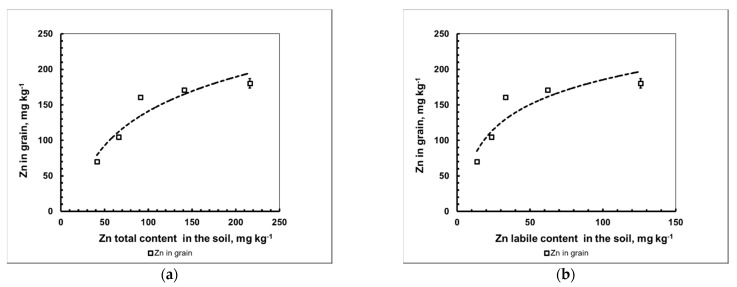
Dependence of the mass concentration of Zn in barley grain on the total amount (**a**) and labile form content (**b**) of the metal in the soil (mean value ± standard deviation; *n* = 3).

**Figure 4 plants-10-02496-f004:**
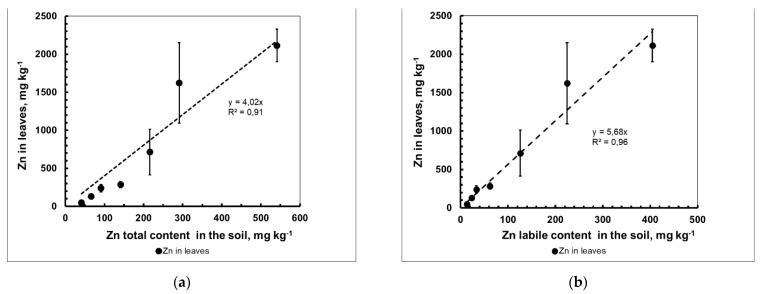
Dependence of the mass concentration of Zn in the phytomass of 21-day-old barley plants on the total amount (**a**) and labile form (**b**) of the metal in the soil (mean value ± standard deviation; *n* = 3).

**Figure 5 plants-10-02496-f005:**
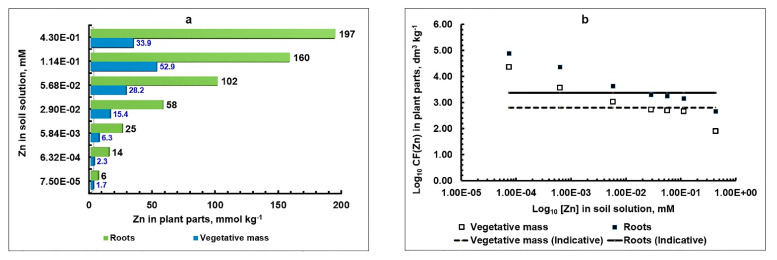
The relationship between the concentration of Zn in the soil solution and its content in different parts of 21-day-old barley plants (in terms of dry biomass) (**a**); values of *CF_Zn,roots_* and *CF_Zn,_*_VPs_, expressed for clarity in logarithmic form (**b**). Straight lines denote constant values in the “indicative” region *CF_Zn,roots_* = 2375 and *CF_Zn,_*_VPs_ = 640 dm^3^ kg^−1^.

**Figure 6 plants-10-02496-f006:**
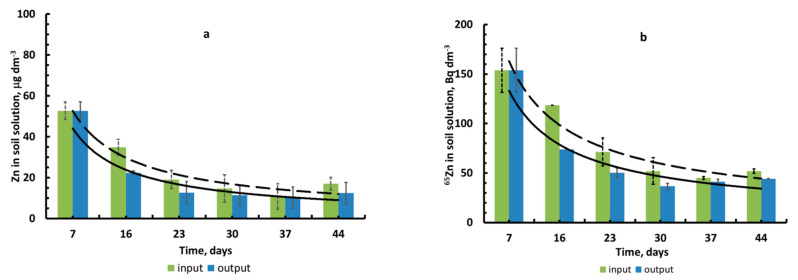
Experimental data for the soil solution in [Zn]_stable_ in µg dm^−3^ (**a**); volumetric activity density (^65^Zn) in Bq dm^−3^ (**b**) (mean value ± standard deviation; *n* = 3).

**Figure 7 plants-10-02496-f007:**
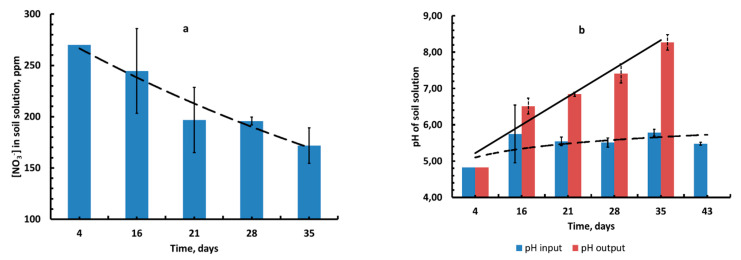
Concentration dynamics in soil solution (mean value ± standard deviation; *n* = 3): [NO_3_^−^] (**a**) and pH (**b**). Data on the 4th day were derived from single extractions.

**Figure 8 plants-10-02496-f008:**
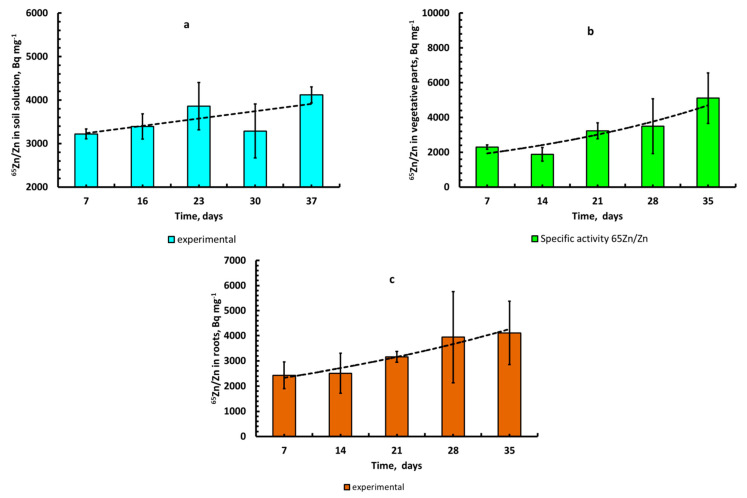
Experimental data for the specific activity: A_sp_(^65^Zn/Zn)_soil solution_ (**a**), A_sp_(^65^Zn/Zn)_VPs_ (**b**), and A_sp_(^65^Zn/Zn)_roots_ (**c**), Bq×mg^−1^ (mean value ± standard deviation; *n* = 3).

**Figure 9 plants-10-02496-f009:**
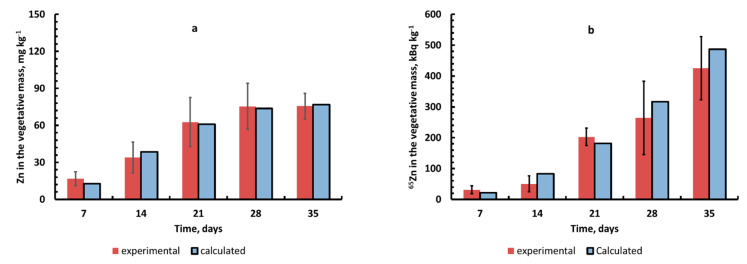
Dynamics of Zn concentration in mg kg^−1^ (**a**); mass activity concentration of ^65^Zn (**b**) (mean values ± standard deviation; *n* = 3).

**Figure 10 plants-10-02496-f010:**
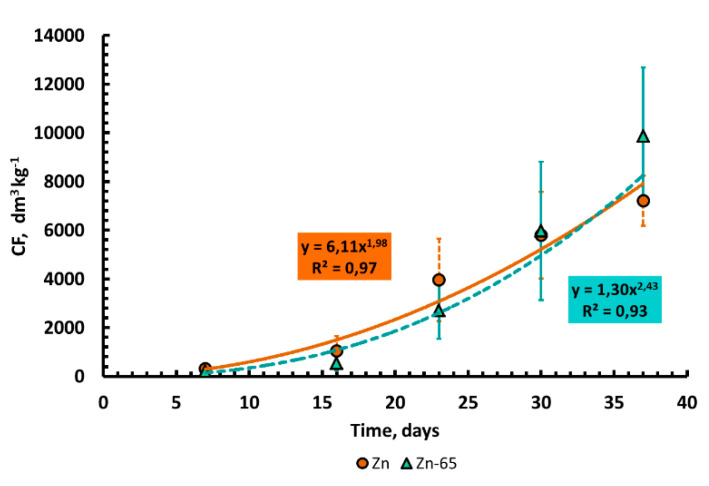
Dynamics of Zn(^65^Zn) concentration factors: *CF*_Zn_, and *CF*_Zn-65_, dm^3^ × kg^−1^ in the dry vegetative mass of barley (mean values ± standard deviation; *n* = 3).

**Figure 11 plants-10-02496-f011:**
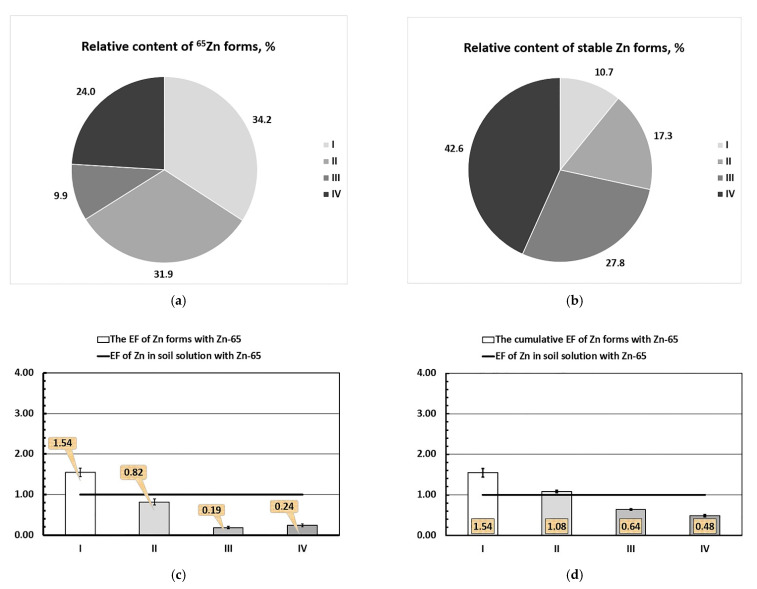
The relative content (%) of the forms of stable (natural) Zn (**a**) and ^65^Zn (**b**) in Albic Retisol (Loamic, Ochric) soil, as determined by the modified BCR method [21]; the enrichment factors (*EF*) of Zn forms by ^65^Zn in the soil (**c**); the cumulative EF of Zn forms by ^65^Zn in the soil (**d**). Roman numerals I–IV indicate the forms of zinc in the soil, as given in Table 2.

**Figure 12 plants-10-02496-f012:**
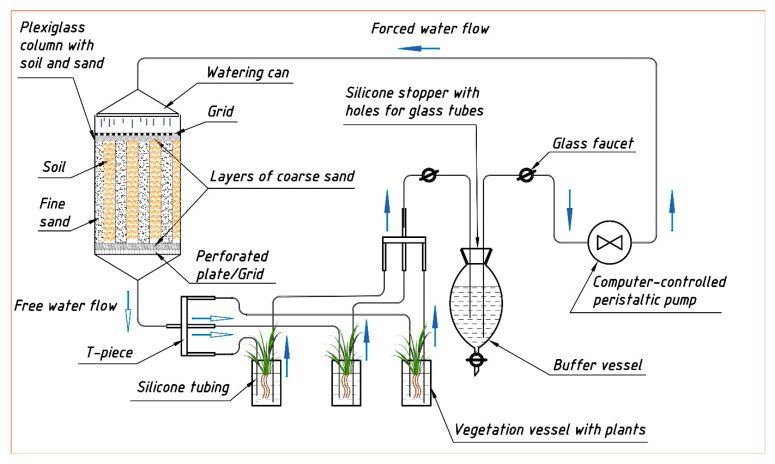
Structural and dynamic scheme of the stand used for studying the parameters of Zn(^65^Zn) migration in the soil–soil solution–plant system (opened blue arrow is designated to soil solution before vegetation vessels (input), filled blue arrow—after them (output)).

**Table 1 plants-10-02496-t001:** Main characteristics of Albic Retisol (Loamic, Ochric) soil (mean ± standard deviation).

Parameter	Value
Mass fraction of particles (mm) in soil, %	
1–0.25	35.08
0.25–0.05	15.64
0.05–0.01	30.88
0.01–0.005	5.20
0.005–0.001	7.30
<0.002	8.75
<0.001	5.89
Exchangeable cation content, cmol (+) kg^−1^	
Ca^2+^	5.20 ± 0.06
Mg^2+^	0.40 ± 0.09
K^+^	0.15 ± 0.01
pH_KCl_	5.05 ± 0.01
pH_water_	6.04 ± 0.01
C_org_, %	1.0 ± 0.01
Total acidity (TA), cmol (+) kg^−1^ soil	1.89 ± 0.02
Total exchangeable bases (S), cmol (+) kg^−1^	5.3 ± 0.2
Labile P_2_O_5_, mg kg^−1^ (Kirsanov method)	126.9 ± 1.9
Mass fraction of total Zn in native soil, mg kg^−1^	37.1 ± 2.8

**Table 2 plants-10-02496-t002:** The determined values of the parameters used to calculate *PBC(V)*_Zn_ (according to experiments 1 and 2).

Parameter	Value
**Experiment I**	
[Ca + Mg]_VPs_, mmol kg^−1^	526 ± 9
[Ca + Mg]_exch_, mmol kg^−1^	23.7 ± 4.4
*CR* _(Ca+Mg), exch_	22.2
*CR* _Zn, exch_	5.68
**Experiment II**	
[Ca + Mg]_VPs_, mmol kg^−1^	613 ± 197
[Ca^2+^ + Mg^2+^]_soil solution_, mM	1.81 ± 0.26
*CF*_(Ca+Mg)_, dm^3^ kg ^−1^	339
*CF*_Zn_ dm^3^ kg^−1^	640

**Table 3 plants-10-02496-t003:** Sequental Extraction Procedure by BCR modified method [21].

*#Form (Chemical Fraction of Zn(^65^Zn)*/Extraction with	Procedure
**I.***Exchangeable and carbonate bound*/Acetic acid, 0.11 M	**I.** A sample of raw soil of known humidity (corresponding to 1 g of absolutely dry soil) without signs of gluing was placed in a 50 mL centrifuge tube. Then, 40 cm^3^ of Solution A were added, the tube was closed with a lid, and the material was extracted by shaking for 16 h at 22.5 °C (or overnight) on a rotator. There was no delay between the addition of the extractant solution and the start of shaking. Then, the extract was separated from the solid precipitate by centrifugation at 3000 g for 20 min and the subsequent decantation of the supernatant into a volumetric glass flask (V = 100 mL) with a polished stopper. Next, 20 cm^3^ of deionized water were added to the sediment, which was shaken for 15 min on a reciprocating shaker and centrifuged for 20 min at 3000 g, and the washing waters were separated by decantation and combined with the extract in a measuring flask. The solution in the flask was brought to the mark with deionized water, stirred, filtered through a 0.45 microns membrane filter, and analyzed for the content of Zn(^65^Zn).
**II.***Associated with reducible Fe–Mn oxides*/Hydroxylammonium Chloride (Hydroxylamine Hydrochloride), 0.5 M (pH 1.5, HNO_3_, 2 M fixed vol.)	**II.** We added 40 cm^3^ of freshly prepared Solution B to the remaining soil after stage (**I**) in a centrifuge tube (see above). The contents were mixed, achieving the complete dispersion of the residue by manual shaking. The centrifuge tube was closed with a lid, and the studied elements were extracted from the soil by mechanical shaking for 16 h at 22.5 °C (night). There was no delay between the addition of the extractant solution and the start of shaking. The procedure for separating the extract, washing the sediment, and preparing the analyte sample was performed in the same way as in step (**I**). It was necessary to carefully ensure that during the last operation we did not accidentally lose part of the solid residue.
**III.***Associated with oxidizable organic matter and sulfides*/Solutions C and D. Solution C: Hydrogen peroxide, 300 mg g^−1^, i.e., 8.8 M, stabilized HNO_3_ to pH 2–3. Solution D: Ammonium acetate, 1.0 M, adjusted to pH 2.0 with HNO_3_.	**III.** We carefully added 10 cm^3^ of Solution C (in small aliquots to avoid losses due to a violent reaction) to the remainder of the soil in a centrifuge tube after stage (**II**). Then, we covered the tube with a lid (loosely) and kept it for 1 h at room temperature (shaking by hand periodically) to oxidize the organic components of the soil with hydrogen peroxide. Then, the oxidation was continued for another 1 h at 85 ± 2 °C in a water bath; during the first ½ hour, centrifuge tubes with soil and extraction solution were periodically manually shaken. The volume of the contents in the test tube with the lid removed was evaporated to approximately V < 3 cm^3^. Then, aliquots of Solution C with a volume of 10 cm^3^ were repeatedly added to the contents of the centrifuge tube. We covered the tube with a lid (leaky) and again continued the oxidation of its contents for another 1 h at 85 ± 2 °C, periodically manually shaking the centrifuge tubes for the first ½ hour. Then, we removed the lid and evaporated the liquid in the test tube to about V ≈ 1 cm^3^, thus preventing the complete drying of the sample. We added 50 mL of Solution D to the cooled wet residue in the test tube and shook it for 16 h at a temperature of 22 ± 5 °C (or overnight). There was delay between the addition of the extractant solution and the start of shaking. The procedure for separating the extract, washing the sediment, and preparing the analysis sample was performed in the same way as in step (**I**).
***Solution A***. In a fume cupboard, we added 25 ± 0.2 cm^3^ of glacial acetic acid to about 0.5 dm^3^ of distilled water in a 1 dm^3^ graduated polypropylene or polyethylene bottle and made up to 1 dm^3^ with distilled water. We took 250 cm^3^ of this solution (acetic acid, 0.43 M) and diluted it to 1 dm^3^ with distilled water to obtain an acetic acid solution of 0.11 M. ***Solution B***. We dissolved 34.75 g of hydroxylammonium chloride in 400 cm^3^ of distilled water. We transferred the solution to a 1 L volumetric flask, and added 25 cm^3^ of 2 M HNO_3_ (prepared by weighing from a suitable concentrated solution) by means of a volumetric pipette. We made up to 1 dm^3^ with distilled water. We prepared this solution on the same day the extraction was carried out. ***Solution C***. 8.8 M water solution H_2_O_2_ (comprised 300 mg g^−1^ of hydrogen peroxide), was stabilized with HNO_3_ to pH 2–3. It is recommended to use hydrogen peroxide acid-stabilized by the manufacturer to pH 2–3. ***Solution D***. We dissolved 77.08 g of ammonium acetate in 800 mL of distilled water and adjust the pH to 2.0 ± 0.1 with concentrated HNO_3_ and made up to 1 L with distilled water.	

## Data Availability

Data sharing is not applicable to this article.

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
