# Peer review of "Zinc Plant Uptake as Result of Edaphic Factors Acting"

_plants, 2021, doi:10.3390/plants10112496_

Round 1

Reviewer 1 Report

An excellent and timely contribution. The use of isotope techniques are making a significant contribution to our understanding of the chemical  aspects of soil-plant relations .

Minor editing to t-remove a few words eg line 43 - leader? an example?

line 43of finding ? delete?

line 66"correct"??  reproducible or more accurate?

lines 90-part 91 ??

line 127 correctness?

Careful editing.

 I found confusing  information on Materials And Methods- seems like some information provided at the beginning of Results and Discussion while most described later on pages 18to 24? Please review!

Author Response

Dear Editor,

We would like to extend our grateful thanks to the reviewer for constructive comments and suggestions, which are helpful for improving our manuscript. We have tried our best to revise our manuscript according to the comments and suggestions from the reviewers. Please find our point-by-point response to the reviewers’ comments below.

Response to Reviewer 1 Comments

Minor editing to t-remove a few words eg line 43 - leader? an example?

line 43of finding ? delete?

Response 1: Sorry for the mistakes. We have changed the part of the sentence “…is the leader in the degree of prevalence…” and abbreviation “TM” in the revised manuscript in Lines 43-44.

line 66"correct"??  reproducible or more accurate?

Response 2: Thank you for your suggestion. We used expression “more accurate” in the revised manuscript in Line 67

lines 90-part 91 ??

Response 3: Sorry for the mistakes. We have translated the untranslated part of the sentence into English in the revised manuscript in Lines 91-93.

line 127 correctness?

Response 4: Thank you for the reviewer's comment. We used expression “validity” instead of “correctness” in the revised manuscript in Line 127.

Careful editing.

 I found confusing  information on Materials And Methods- seems like some information provided at the beginning of Results and Discussion while most described later on pages 18to 24? Please review!

Response 5: We have moved some of the material from the "Results and discussion" section to the "Materials and Methods" section (in particular, Table 2). In addition, we have restructured the material in the "Materials and Methods" section

Reviewer 2 Report

Review of the article: “Zinc Plant Uptake as Result of Edaphic Factors Action”

It was a very interesting article for me. The authors used in the research

  labile zinc compounds. The methods used and the scope of the analyzes allowed to achieve the aim of the research.

I found only a few editing errors in the manuscript

Since there was one sentence in the text written with cyrillic letters (P2 L 90), I suggest that the publication be revised linguistically by a native speaker

P 18 L 599 change to:  Zn(NO3)2

P 24 L 877 and P 25 L 900 remove green shaded

I found no more editing errors. The literature was well cited.

Author Response

Response to Reviewer 2 Comments

Dear Editor,

We would like to extend our grateful thanks to the reviewer for constructive comments and suggestions, which are helpful for improving our manuscript.

Review of the article: “Zinc Plant Uptake as Result of Edaphic Factors Action”

It was a very interesting article for me. The authors used in the research

  labile zinc compounds. The methods used and the scope of the analyzes allowed to achieve the aim of the research.

I found only a few editing errors in the manuscript

Since there was one sentence in the text written with cyrillic letters (P2 L 90), I suggest that the publication be revised linguistically by a native speaker

Response 1: Sorry for the mistakes. We have translated the untranslated part of the sentence into English in the revised manuscript in Lines 91-93.

P 18 L 599 change to:  Zn(NO3)2

Response 2: Thank you for the reviewer's comment. We have made a correction in the zinc nitrate formula in the revised manuscript in Line 599.

P 24 L 877 and P 25 L 900 remove green shaded

Response 3: Thank you for the reviewer's comment. We have made a correction in Line 900 of the revised manuscript.

I found no more editing errors. The literature was well cited.

Reviewer 3 Report

This paper has high reference value for the utilization of zinc fertilizer. However, some contents need to be added and appropriate improvements need to be made before publication.

1, Why is Russian on line 90?

2, The table on page 10 is not numbered.

3, The table on page 11 should be changed to a three line table.

4, Why is there no error line on four days in Figure 7?

Author Response

Response to Reviewer 3 Comments

Dear Editor,

We would like to extend our grateful thanks to the reviewer for constructive comments and suggestions, which are helpful for improving our manuscript.

This paper has high reference value for the utilization of zinc fertilizer. However, some contents need to be added and appropriate improvements need to be made before publication.

1, Why is Russian on line 90?

Response 1: Sorry for the mistakes. We have translated the untranslated part of the sentence into English in the revised manuscript in Lines 91-93.

2, The table on page 10 is not numbered.

Response 2: Thank you for the valuable comment. We have canceled the presentation of a sequential series of formulas in the form of a table

3, The table on page 11 should be changed to a three line table.

Response 3: Thank you for the valuable comment. We have changed this table to a three line table

4, Why is there no error line on four days in Figure 7?

Response 3: Thank you for the valuable comment. Unfortunately, it so happened that the pH and composition of soil solutions (including nitrates) at the very beginning of the vegetation experiment - on the 4th day, we monitored only once a day - separately at the input and output from the vegetation vessels. Figure 7 shows the average value for nitrates (270 = (260+280)/2). pH values at the input and output of vegetative vessels were 4.82 and 4.83, respectively. On the remaining days coinciding with the gathering of plants, the parameters of soil solutions were monitored three times a day - in the morning, afternoon and evening. In Fig. 7, we have added an appropriate explanation: “Data on the 4-th day are single”